# Kid Growth Comparison between Murciano-Granadina and Crossbred Murciano-Granadina×Boer in a Mixed Rearing System

**DOI:** 10.3390/ani11041051

**Published:** 2021-04-08

**Authors:** Nemesio Fernández, José L. Palomares, Ión Pérez-Baena, Martín Rodríguez, Cristòfol J. Peris

**Affiliations:** Instituto de Ciencia y Tecnología Animal, Universitat Politècnica de València, Camino de Vera s/n, 46022 València, Spain; jopalom@dca.upv.es (J.L.P.); iopebae@upv.es (I.P.-B.); mrodriguez@dca.upv.es (M.R.); cperis@dca.upv.es (C.J.P.)

**Keywords:** mixed rearing system, Murciano-Granadina, crossbreeding, kids, profitability

## Abstract

**Simple Summary:**

Usually, goat dairy farms have a major problem with the kids they produce for sale, due to their low average daily gain, high feed conversion ratio and poor body conformation, which leads to poor profitability. This paper proposes a solution by crossing the dams not used for replacement with males of a meat breed in a mixed rearing system. The results show that crossbred kids (Murciano-Granadina×Boer) reached the minimum slaughter weight a week earlier than Murciano-Granadina purebred kids, and that there is a decrease of energy content in milk and in the consumption by the kids as their age increases, which shows the importance of having a concentrated feed that complements their needs to express all the genetic potential for growth observed in the references, especially in the crossing of these two breeds. In addition to this, farms that follow this strategy will also be able to improve their profitability through a higher quantity of milk sold.

**Abstract:**

In dairy goats, the low average daily gain and the high conversion ratio for milk and concentrate of the kids mean that their sale price does not offset the costs generated. The hypothesis proposes that a crossbreeding of the Murciano-Granadina breed (MG) with the Boer breed (MG×Boer) will improve the profitability of the kids sold. Thus, the effect of two different groups of kids (purebred MG and crossbred MG×Boer) on birth weight (BW), mortality, average daily gain (ADG), the time for minimum slaughter weight (7 kg) and its variation factors were studied. MG×Boer kids had a 27% greater BW than purebred MG kids (2885 ± 84 g and 2275 ± 74 g, respectively), similar ADG (156 ± 6 g and 142 ± 6 g, respectively) and mortality (18% and 20%, respectively), and reached minimum slaughter weight a week sooner. ADG was less and less as the lactation period progressed due to a lower milk consumption and milk energy value, which highlights the importance of providing a concentrate that will compensate for this reduced energy content. In conclusion, the results show that MG×Boer crossbred kids reached the minimum slaughter weight a week earlier than purebred MG kids, and highlighted the improvement of farm profitability through the increase of milk sold and the need to provide a concentrate feed to enhance the growth of the kids.

## 1. Introduction

In a dairy goat farm, the amount of commercial milk and the growth of kids depend, among other factors, on the rearing system and the age at weaning [1]. The artificial rearing system (ARS), exclusively with milking from parturition, and the mixed rearing system (MRS), which includes a suckling plus milking period before weaning, are the two main systems for the production of goat milk in Spain [2,3]. On the one hand, the intensification of the farms, the increase in the price of milk and the seasonality and low prices of kid meat, in addition to the complication of having to separate the mothers at milking time, are driving the change from MRS to ARS [4,5]. On the other hand, ARS requires high investments in facilities and machinery [3], which leads to an increase in costs [6], and thus, interest in MRS is maintained. Moreover, the average daily gain (ADG) of the Murciano-Granadina (MG) kids is very low (120–168 g/day) [7,8] compared to other Spanish goat breeds such as Florida (184–203 g/day) [9] or Verata (164–180 g/day) [10] or to foreign goat breeds such as Damascus (180 g/day) [11] or Boer (>200 g/day) [12,13]. Additionally, purebred MG kids have high feed conversion rates in ARS (1.55 kg/kg) [7], which compromises the profitability of their production. A possible solution could be to mate females that are not used to produce the next generation of replacements with males of a meat breed. It is hypothesised that the crossbreeding between MG females and Boer males, in a mixed rearing system, will improve the ADG of the kids, which will reduce the time necessary to reach the minimum commercial slaughter weight. In addition, some of the factors (genetic, sex and number of reared kids) that affect these variables were studied.

## 2. Materials and Methods

### 2.1. Ethics

Housing and handling of the experimental animals followed the mandatory principles for care and use of experimental animals in Spain (Real Decreto 53/2013, Boletín Oficial del Estado 34, 11370-11421).

### 2.2. Goats and General Procedures

Eighty-two multiparous (2.8 ± 0.3 years) MG breed goats weighing on average 44 ± 2 kg were used at the experimental farm of the Universitat Politècnica de València (Spain). Half of the does were mated with five MG males and the other half with five Boer males. In each group of 41 goats, the presentation of the heats was programmed in two days (20 and 21 does per day) through hormonal treatment, and carrying out a new mating at the arrival of the second heat 19–21 days later. Mating was synchronised by intravaginal sponges (30 mg fluorogestone acetate, Chrono-gest^®^, CEVA Salud Animal, Barcelona, Spain) and injected with 450 International Units (IU; PMSG, CEVA Salud Animal, Barcelona, Spain), and all births took place over a 23-day period. Of the initial 82 goats, 16 goats (10 crossed with MG males and 6 with Boer males) were removed from the experiment due to the death of their only kid (4 goats) or of one or both of their two kids (12 goats). Finally, the experiment involved 32 goats that raised one kid each and 34 goats that raised two kids each. The distribution of kids by sex between the genetic types was 26 females (11 from single births and 15 from double births) and 19 males (6 from single births and 13 from double births) for purebred MG kids, and 31 females (8 from single births and 23 from double births) and 24 males (7 from single births and 17 from double births) for crossbred MG×Boer kids. Kids suckled freely and goats were subjected to once-a-day milking (0800 h) for the entire lactating period studied (0–6 weeks postpartum), and with minimal care by caregivers. All goats were kept together in the same pen (size = 1.5 m^2^/goat; feeder = 0.5 m/goat; five bowl water troughs) and received the same total mixed ration twice daily (0900 h and 1800 h). The ration was formulated according to Sauvant et al. [14] and consisted of: (1) a basal diet to meet minimum recommendations for maintenance plus 1.5 L milk/day (8.71 MJ net energy; 99 g metabolisable protein; 8.7 g Ca and 4.9 g P per kg DM) including alfalfa hay (30% as DM), barley straw (26%), beetroot pulp (18%), orange pulp (26%), and (2) a commercial concentrate for dairy goats (6.78 MJ net energy, 135 g metabolisable protein, 9 g Ca and 4 g P per kg of DM) to meet a total average milk yield of 3.2 L milk per goat per day. Rations were offered to the dams in an amount 10% higher than the calculated voluntary feed intake. Throughout the experimental period, pens and feeders were arranged so that the kids had no access to the feed provided to the dams, and therefore, the only source of nutrients available to the kids was maternal milk. A medium line milking parlour (two platforms; 12 goats per platform; six milking units) was used; machine milking parameters were set to: vacuum = 42 kilopascals, pulsator rate = 120 cycles per minute and pulsator ratio = 60%. Kids were weighed using a digital dynamometer (KERN^®^ HDB 10KG10, Masnou, Spain) at birth and weekly thereafter until weaning from their dams. When necessary, individual weight was adjusted to a determinate day of a week taking into account its own weekly ADG.

### 2.3. Experimental Data and Sample Collection

Mortality rate takes into account the dead kids during the experiment but does not include those that had died at parturition. Actual milk yield (milked milk) was recorded once a week at 0800 h on Tuesday. Immediately afterwards, potential milk yield was determined according to the double oxytocin injection method [15,16]. Goats were injected twice with 3 IU of oxytocin (Hormonipra^®^; Laboratorios Hipra, Gijón, Spain) into the jugular vein, with a 4-h time interval between injections. The udder was emptied by machine and the milk volume obtained after the second injection was multiplied by 6 to estimate the daily potential milk yield. Milk samples (50 mL) were collected and immediately analysed for milk composition and Somatic cell count (SCC). Milk composition (fat and protein) was analysed with an infrared analyser (Milkoscan^®^ FT6000; Foss Iberia, Barcelona, Spain) and SCC was determined by the fluoro-opto-electronic method (ISO, 2008; Fossomatic^®^ 5000, Foss Iberia, Barcelona, Spain). The energy content (EC) of milk was estimated according to Daza et al. [17]:EC (Kj/kg) = 38.00 (kj/g) × F + 24.44 (kj/g) × P + 16.45 (kj/g) × L
where:

G = fat content in milk (g/kg),

P = protein content in milk (g/kg),

L = lactose content of milk (g/kg).

### 2.4. Weighing-Suckling-Weighing Milk Yield Consumed Estimation

Daily milk consumed by the kids was measured by the weighing-suckling-weighing method (WSW) weekly, on Thursday. After machine milking (0800 h), the dams returned to the pens, where they remained nearby but separated from the kids for a 4-h period to prevent suckling. Following this separation period, the kids were weighed to the nearest 10 g and allowed to suckle from their mothers for 5 min, and weighed again to evaluate the milk yield consumed. This process was repeated for each 4-h period during 24 h on the experimental days. Daily milk consumed by kids was estimated by the sum of milk yield obtained by the WSW method during 24 h.

### 2.5. Statistical Analysis of Results

Weekly evolution of actual milk, milk consumed by the kids, potential milk energy, SCC and ADG of the kids were statistically analysed with a repeated measures model. This mixed model included the fixed effects of genetic type (purebred MG kids or crossbred MG×Boer kids) and sex of the kid, the number of kids reared (1,2), week of lactation (1–6) and the double interactions among these four factors, the random effect of animal and residual error. When an interaction was non-significant (*p* > 0.05), the corresponding interaction term was pooled with the error. This model was analysed using the MIXED procedure in SAS [18]. The SCC logarithm (SCClog) was used to normalise SCC distribution [19]. Mean values of the ADG were analysed statistically using a model that included the fixed effect of genetic type and sex of the kids and number of kids reared, and residual error (Proc. GLM SAS) [18]. For all models, separation of the means for the determination of a significant (*p* < 0.05) main effect was performed using pairwise contrasts (PDIFF option from SAS) [18]. Kid mortality was analysed statistically using Proc. Freq [18], while correlations were analysed by Proc. Corr. [18].

## 3. Results

### 3.1. Kid Mortality

Out of a total of 124 kids, there were 24 deaths (19%) throughout the experiment. The BW (*p* = 0.002) and the number of kids reared (*p* = 0.010) had a significant effect on kid mortality, the genetic type was on the limit of significance (*p* = 0.072; 21% and 18% of mortality for purebred MG and crossbred MG×Boer, respectively) and the sex of the kid was not significant (*p* = 0.364; females = 20%, males = 18%). Table 1 shows that the percentage of kid mortality decreased when the BW was higher, and that from 2500 g BW the mortality remained low and constant. Thus, 100% of the kids with BW under 1500 g died during the experiment, as well as 48% of the kids under 2000 g.

### 3.2. Kid Growth

Figure 1 shows the evolution of the kids’ live weight until the sixth week of lactation, depending on their genetic type. Birth weight of crossbred MG×Boer kids (2885 ± 84 g) was 26.8% higher (*p* < 0.01) than that of MG kids (2275 ± 74 g). The superiority of the BW of the crossbred kids compared to the pure MG (610 g) was increased to 920 g at the end of the six weeks of the lactation period (8830 ± 240 g vs. 7910 ± 199 g), so that the BW was positively correlated (r = 0.556, *p* < 0.0001) with the final weight. Average daily gain was 149 g/day for the total of 100 surviving kids, a value that was significantly affected by the sex of kids (Table 2), although not by their genetic type or the number of kids reared. Thus, males grew 23 g/day more than the females. In addition, Figure 2 represents the ADG evolution for the genetic type and sex of the kids, showing that this trend was decreasing as the individuals got older. The difference in ADG between genetic types was greater at the beginning of the lactation period than at the end of it.

### 3.3. Production, Consumption and Characteristics of Milk

The quantity of actual milk was affected significantly by the number of kids reared (*p* = 0.002), by the week of lactation (*p* < 0.0001) and by the sex of the kid (*p* = 0.033), but not by the genetic type (*p* = 0.786), and there was a significant interaction between the number of kids reared*week of lactation (*p* < 0.0001). In turn, the milk consumed by the kids presented significant differences depending on the number of kids reared (*p* < 0.001) and the week of lactation (*p* = 0.004), and genetic type factor was very close to significance (*p* = 0.053), while potential milk depended only on the week of lactation (*p* < 0.001).

Table 3 shows that actual milk was much higher (700–900 g/day) in the first four weeks of lactation for the goats that reared a single kid compared to those that raised two, while in the last two weeks of lactation, the amount of actual milk was similar (60–180 g/day of difference). The average consumption of milk decreased as the age of the kids increased (Table 4). This trend in milk consumed was consistent irrespective of the number of reared kids and the genetic type of the kids. In turn, the amount of potential milk tended to increase between the first and sixth week of lactation (from 2532 ± 121 to 3000 ± 125 g/day, respectively).

Fat and protein milk composition (*p* < 0.001), the energy of the milk (*p* = 0.001) and the SCClog (*p* = 0.027) changed significantly based on the stage of lactation and, for SCClog, also on the interaction between number of kids reared*lactation week (*p* = 0.037). Average values of the potential milk energy (Table 5) fluctuated between 3574 and 3772 kj/kg for the first two weeks of lactation, while they did not exceed 3441 kj/kg in the rest of the weeks of the experimental period, similar to what happens with milk fat and protein, with those that presented positive and significant correlations (r = 0.96 *p* < 0.001 and r = 0.78 *p* < 0.001, respectively). In turn, the logarithm of the SCC presented mean values of 5.6 ± 0.01.

## 4. Discussion

The mortality rate of kids in this paper was 21%. This value could be reduced through greater attention to the kids by the farmer, as in this experiment minimal attention was paid to determining how BW affects mortality. Kids with BW in the 2001–2500 g range showed a mortality rate of 14%, which is of the same order as that obtained (16%) by Pérez-Razo et al. [20] for the same breed and a similar range, while mortality above 2500 g was very low (3–4%).

The BW values obtained in this work are similar to those found by Pérez-Baena et al. [7] between animals with the same genetic type (MG = 2180 g and MG×Boer = 2790 g). However, the BW of MG kids from the current study is lower than that obtained, in the same breed, by Fuentes et al. [21] (2440 g), Sanz [22] (2700 g), Vazquez-Briz et al. [8] (2430 g) and Argüello et al. [23] (3005–3350 g) in the Canary caprine group. In turn, the BWs of crossbred kids are similar than those reported by Pérez-Baena et al. [24] (2790 g) for the same breeds and lighter than those reported by Goonerwardene et al. [25], who also evaluated the BW of crossbred kids from Boer sires and dairy breed dams, such as Alpine (3950 g) and Saanen (4090 g), although both breeds have an adult live weight higher than the MG breed. However, all of the BWs discussed are lower than those of the purebred Boer (3000–4000 g), reviewed by Van Niekerk and Casey [26] and by Lu [27].

On the other hand, Pérez-Baena et al. [7] observed that MG×Boer crossbred kids presented an ADG (163 g/day) greater (*p* < 0.001) than that of the purebred MG (128 g/day) in ad libitum ARS. These results were slightly greater than those obtained in this work for crossbred kids (156 g/day), and lower than the ones for purebred MG (142 g/day). Moreover, in this study, the ADG reduced as the individuals got older, although this trend was not observed in other studies [7,8]. For example, Vazquez-Briz et al. [8] showed ADG values of pure MG kids, in an ARS, of 105, 192, 169, 150 and 153 g/day, for the first, second, third, fourth and fifth weeks of life, respectively, and, in MRS, of 155, 222, 156, 147 and 180 g/day, for the same weeks, respectively. Something similar occurs with the results of Pérez-Baena et al. [7] for pure MG and crossbred MG×Boer, in an ARS. In fact, the results of this work go against the theoretical higher milk consumption capacity of the kids when their live weight is increased, given that the live weight of the kid was positively correlated (r = 0.409, *p* = 0.001) to the milk consumption. A possible explanation would be that mothers, as their kids grew, were less likely to allow them to suckle due to, among other possible causes, the appearance of wounds on the teats, injuries that were observed at milking time. In addition, this behaviour of the mothers was also observed in the WSW records used to evaluate the amount of milk consumed by the kids. These observations highlight the importance of the kids having a concentrated feed that can compensate for a lower hypothetical milk consumption, since, in this experiment, the kids were not allowed to access any type of food other than maternal milk. On the other hand, the decrease in the energy value of milk after the second week of the lactation period, together with the decrease in the amount of milk consumed, explains the decrease in ADG mentioned above. The results of higher ADG at 35 g/day for MG×Boer kids compared to pure MGs, observed by Pérez-Baena et al. [7] in an ad libitum ARS, compared to the 14 g/day obtained in this work, could suggest that a lower consumption of milk and, in addition, of lower energy value, would have prevented the kids manifesting their full growth potential. The average weight of 35 days old purebred MG kids (7293 g) was similar to the 7400 g reported by Zurita-Herrera et al. [28], greater than in Pérez-Baena et al. [7] (6560 g), but lower than that of Vazquez-Briz et al. [8] (8402 g). In Spain, the typical average slaughter live weight for kids is around 7000 g [29]. In this experiment, purebred MG kids reached 7000 g at 32 days, while crossbred kids can achieve slaughter weight one week earlier, allowing an increased in the quantity of milk sold and improving the economic results of the farm.

To give an idea of the conditions in which this experiment was carried out, the fact that the SCClog average value in milk was between 5 and 6 points to a good sanitary state, in general, of the udders, given that values of 6 and 6.1 of tank milk in MG commercial farms are common [30].

## 5. Conclusions

This work compared the growth of purebred MG kids with those obtained in the crossing of MG females with Boer males. The latter had a birth weight greater than that of purebred MG kids, and reached the minimum slaughter weight typical in Spain (7000 g) a week earlier, allowing an increase in the quantity of milk sold, and thus, improving the farm profitability. There is a decrease in milk consumption by the kids as their age increases, which shows the importance of having a concentrated feed that complements their needs in order to express all their genetic potential for growth.

## Figures and Tables

**Figure 1 animals-11-01051-f001:**
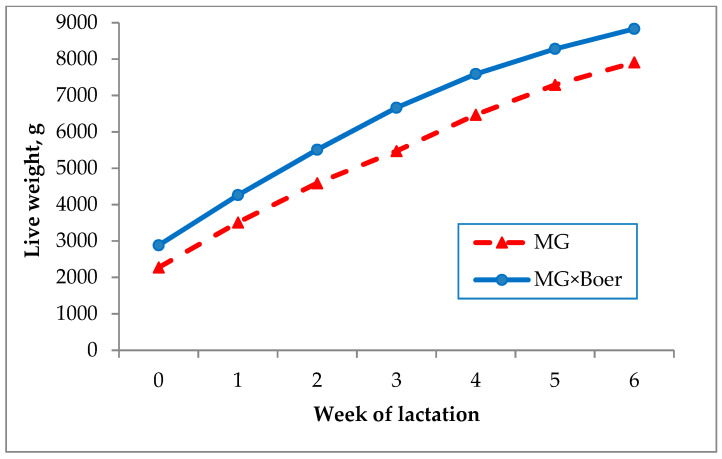
Evolution of unadjusted kids’ live weight according to their genetic type, in a mixed rearing system (MG = Murciano-Granadina; week 0 corresponds to the birth weight).

**Figure 2 animals-11-01051-f002:**
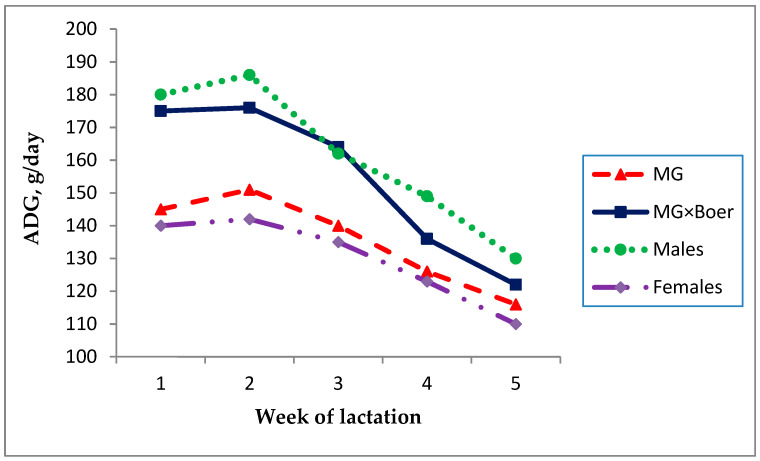
Evolution of the average daily gain (ADG) associated with the effects of genetic type and sex of the kids, in a mixed rearing system (MG = Murciano-Granadina).

**Table 1 animals-11-01051-t001:** Frequencies of kid mortality according to the weight at birth, in a mixed rearing system.

Birth Weight (g)	Mortality		Total Kids
	Number	Percentage	
<1500	7	100	7
1501–2000	10	48	21
2001–2500	5	14	37
2501–3000	1	3	34
>3000	1	4	25
Total kids	24	19	124

**Table 2 animals-11-01051-t002:** Mean values of the average daily gain associated with the effects of genetic type, sex of the kids and the number of reared kids, in a mixed rearing system (MG = Murciano-Granadina).

Trait (n)	Mean (g/day)	SEM	*p*-Value
Genetic type			
MG (45)	142	6	*p* = 0.082
MG×Boer (55)	156		
Sex			
Male (43)	160	6	*p* = 0.006
Female (57)	137		
Reared kids			
1 (32)	147	6	*p* = 0.679
2 (68)	150		

**Table 3 animals-11-01051-t003:** Actual milk mean values (mL) according to the number of kids reared (1,2) and the week of lactation, in a mixed rearing system.

Week of Lactation	Reared Kids		SEM	*p*-Value
	1	2		
1	1638	764	144	*p* < 0.001
2	1563	773	143	*p* < 0.001
3	1756	972	146	*p* < 0.001
4	1701	1058	146	*p* = 0.002
5	2091	1911	109	*p* = 0.392
6	2212	2015	150	*p* = 0.991

**Table 4 animals-11-01051-t004:** Consumed milk mean values (ml) according to the number of kids reared (1,2) and the genetic type, in a mixed rearing system (MG = Murciano-Granadina; B = Boer).

Week of Lactation	Reared Kids		SEM	*p*-Value	Genetic Type		SEM	*p*-Value
	1	2			MG	MG×B		
1	962	2152	166	*p* < 0.001	1523	1591	118	*p =* 0.658
2	963	2358	168	*p* < 0.001	1621	1701	119	*p =* 0.899
3	927	1979	170	*p* < 0.001	1389	1527	121	*p =* 0.325
4	877	1891	173	*p* < 0.001	1316	1451	124	*p =* 0.284
5	654	1738	174	*p* < 0.001	1110	1282	125	*p =* 0.913
6	644	1554	151	*p* < 0.001	957	1170	130	*p =* 0.614

**Table 5 animals-11-01051-t005:** Average values (kj/kg) of the potential milk energy according to the number of kids reared (1,2), in a mixed rearing system.

Week of Lactation	Reared Kids		SEM	*p*-Value
	1	2		
1	3710	3702	135	*p =* 0.966
2	3772	3574	135	*p =* 0.213
3	3360	3121	136	*p =* 0.241
4	3424	3318	140	*p =* 0.891
5	3282	3239	142	*p =* 0.970
6	3441	3344	145	*p =* 0.871

## Data Availability

Not applicable.

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
