# Peer review of "Kid Growth Comparison between Murciano-Granadina and Crossbred Murciano-Granadina×Boer in a Mixed Rearing System"

_animals, 2021, doi:10.3390/ani11041051_

Round 1
Reviewer 1 Report
Overall, the experiment is well conceived, and the results reported look suggestive, but the quality of the paper needs improvement. Minor suggestions are proposed prior to publication. This reviewer appreciates that the authors discuss different factors that could be influencing the results. I believe that the topic is interesting, and the results are of value to goat farmers and the wider research community. I enjoyed the paper however it was not easy to understand what the author was trying to say. A few sentences did not make sense and the paper requires a bit of re- working.
The aim of the study is clear, and I have no methodological concerns but the clarity and readability of the paper needs work. I have suggested multiple corrections that if taken on board would make it easier for the reader to understand.
Specific comments:
Line 16: How do you know they are not reaching their genetic potential for growth?
Line 20: “the kids mean that their sale price does not compensate the costs generated” -compensate is the wrong word to use here.
Line 25: It is poor English to state things to be higher as higher is referring to a level, instead recommend using other words to explain increase i.e. greater.
Line 25: When discussing breeds it is better to say purebred rather than just pure.
Line 25: doesn’t this equate to 27%
Line 26: remove a after comma “, similar ADG”
Line 26: space after 6 g
Line 27: the ADG was not reduced as the lactation progressed, however the ADG was less and less as the lactation progressed.
Line 28: “and milk energy value” – what is milk energy value?
Line 29: “that compensates said reduction” suggest changing to say “that will compensate for this reduced energy content”.
Line 30: “results show that 29 crossed kids “ be more specific about what you are referring to ... “results show that MGxBoer crossbred kids” and “purebred MG kids”.
Line 30: “In conclusion, the results show that crossed kids reached the minimum slaughter weight a week earlier than pure kids, highlighting the need to provide a concentrated feed and the increased income from higher milk sales”…. The fact that the crossbred animals reached the slaughter weight earlier does not highlight the need to provide a concentrated feed. These are 2 factors, first, the crossbred kids reached the weight fastest, and another point you identified in the study is that the milk energy content is less. Based on these observations, in order to achieve the desired growth these goats should be provided with a concentrated feed. I would suggest you slightly change the language from “a week earlier than pure kids, highlighting the need” to “a week earlier than purebred kids, and highlighted the need”. Also, I don’t see how this highlights the increased income from higher milk sales?
Line 32: key words “breeding system” – only that is not included in the title.
Line 38-39: “which contemplates suckling and milking period postpartum and usually once-a-day milking, are two regular systems for the production of goat milk in Spain [2,3]”… this second system is difficult to understand... Also I don’t think contemplates is the correct word for this.. suggest writing “which includes a suckling period prior to the milking period, which is typically once-a-day milking..”
Line 39: Change “are two regular” to “are the two main systems”
Line 43: On the other hand,
Line 44: “On the other, ARS presents difficulties due to investment in facilities and machinery [3], which leads to an increase in costs [6], and so maintains the interest of the MRS” this sentence doesn’t make sense… please re-write
Line 46: “kids is very low (120-168 g/day)”. Compared to what?
Line 47-48: ” to mate females not used to obtain replacement with males of a meat breed” ... reword “to mate the females that are not used to produce the next generation of replacements, with males of a meat breed.”
Line 50: “average daily gain” …. Be consistent with abbreviations, so change to ADG.. and change “kids and that this will reduce” to “kids which will”
Line 60: “MG breed goats (44 ± 2 kg)” … MG breed goats weighing on average 44 ± 2 kg…
Line 66: raised one kid each?.. two kids each?
Line 74: are the Ca and P per kg of DM too? as with the concentrate?
Line 85-86: and adjusted, when necessary, to 35 day weights… why? And how? This has not been explained previously, why 35day?
Line 88: what is actual milk yield? Was this daily milk yield? Weekly total milk yield? Total milk yield?
Line 94: change to “milk samples (50 ml) were collected..
Line 101: “weighing method (WSW) in every week of the experiment, on Thursday” change to “weighing method (WSW) weekly.”
Line 110: evaluation?.. also “milk consumed by the kids”
Line 111: milk characteristics? Which were?. Average daily gain of the kids or dams? also be consistent with abbreviations “ADG”.
Line 112: “with a repeated measures model. This mixed model included the fixed effects of..”
Line 112: “genetic type” please specify.
Line 113: “week of lactation and their interaction”? What is the week of lactation? Do you mean the week they kidded? Or the number of weeks they have been lactating for? What is the interaction with?
Line 116: was analysed by a mixed model, but what software? Suggest saying “was analysed using the MIXED model procedure in SAS (reff)”.
Line 117: global average daily gain? Was this a global study?
Line 126-127: change “experiment, so that the BW” to “experiment. The BW..”, also, change “were significant on mortality” to ”had a significant effect on kid mortality”
Lines 128-129: not clear what the percentages are...
Lines 130-131: “as the birthweight rose” - the birthweight doesn’t rise, it is one weight for each animal. However, some kids have greater/heavier BWs
Line 132: “Thus, 100% of the kids with BW under “
Line 139: crossbred kids
Line 140: versus change to vs
Line 141: Shouldn’t start sentence with an abbreviation.
Line 142: was this ADG value the average ADG of the 100 kids?
Lines 141-144: big sentence, suggest splitting.
Lines 146: I wouldn’t say the evolution of ADG was negative, as they still grew during this time, the ADG trend however was decreasing as the individuals got older.
Line 146-147: “the difference in ADG between… was greater…”
Line 148: “period than at the end”
Lines 148-149: please re-write as this doesn’t make sense.
Line 164: “and sex factor”. Maybe change wording.
Table 3 – Name does not make sense. Also, it is difficult to understand the table. Are the values under 1 and 2 number of animals or milk yields? Also what does ESM stand for?
Lines 170-171: “On the other hand, Table 4 shows that the average value of milk consumed by the kids evolved negatively as their age increased, a trend that is valid for the number of kids reared and for the genetic type”…. Doesn’t make sense, please reword... “The average consumption of milk decreased as the age of the kids increased (Table 4). This trend in milk consumed was consistent irrespective of the number of reared kids and the genetic type of the kids.”
Table 4- same comments as Table 3
Line 180: the composition doesn’t depend on the stage of lactation, but it does change significantly based on the stage of lactation.
Line 181-182: “in the later case” , do you mean the later stages of lactation?
Line 183: what is milk energy? Never mentioned this before, how have you calculated this?
Lines 183-184: “Table 5 shows how the average values of milk energy tended to decrease from the second week to the sixth week” – if I’m reading the table correctly, the “milk energy” increases up to week 2 then decreased then increased then decreased then increased. …
Table 5- similar comments as Table 3
Line 191: “The level” the proportion? Mortality rate?
Line 191: you compared the mortality to another paper, why could these mortality rates be different? Where was this other study conducted? Same breeds?
Line 192: “the breeding process”? Please explain
Line 194: abbreviation consistency - birth weight = BW
Line 195: What do you mean by habitual mortality?
Line 199: “However, the BW of the MG kids is lower than the one obtained, in the same breed” the BW of the current study MG kids? Or what MG kids?
Line 201: It is better to write crossbred rather than crossed. Also, these values are averages so you should say “ the average BW of crossbred kids” etc.
Line 201: “the BW of crossed kids is lower than the one reviewed by”.. “the BW of crossbred kids are lighter than those reported by Goonerwardene et al. [17], who also evaluated the BW of crossbred kids from Boer sires and dairy breed dams such as Alpine (3950 g) and Saanen (4090 g). Although, ..”
Line 203: “all of them” – who is them?
Line 204: “all these BW” – all of the BWs discussed…
Lines 206-209: please re-word sentence, very hard to follow.
Lines 209-211: “Moreover, descending values of ADG with the progress of lactation observed in this work are contrary to what was reported by other authors.” Suggest re-wording.. “Moreover, in this study the ADG reduced as the individuals got older, however this trend was not observed in other studies (reference the studies). For example, Vazquez...”
Line 217: change raised to increased
Line 219: “when raising the weight of their kids” – what do you mean by this?
Lines 218-220: very confusing sentence, please re-word
Line 222: “these facts” – these observations
Line 223-224: for a hypothetical lower milk consumption” recommend saying lower before hypothetical.
Lines 225: Refer to my comments about lines 183-184
Line 231: why especially the crossed animals?
Line 232: with all this – new sentence so what are you referring to?
Line 232-233: “With all this, the final weight at 35 days of life of this work of pure MG kids (7293 g) was lower than that of Vazquez-Briz et al. [8] (8402 g), similar to that of Zurita-Herrera et al. [20] (7400 g) and higher than Pérez-Baena et al. [7] (6560 g).” suggest re-wording “The average final weight of 35 day old purebred MG kids (7293 g) was similar to the 7400 g reported by Zurita-Herrera et al. [20], greater than Pérez-Baena et al. [7] (6560 g) but lower than that of Vazquez-Briz et al. [8] (8402 g).”
Lines 234-236; also suggest re-wording this sentence to make sense e.g. “In Spain, purebred MG kids reach the minimum typical sales weight at 32 days, while the crossbred kids can achieve slaughter weight one week earlier, allowing an increase in the quantity of milk sold and improving the economic results of the farm.”
Lines 238-241: Do you think these conditions may have had an effect on the results?
Final comment, TSW was only briefly mentioned a few times despite this being one of the main objectives of the study. If you do not refer to this often then I would suggest not abbreviating.
Author Response
RESPONSE TO REVIEWER 1
To make the changes made in the manuscript more visible, these changes are written in red type.
Reviewer 1
Specific comments:
Line 16: How do you know they are not reaching their genetic potential for growth?
Authors: A new sentence has been written. Please see the manuscript.
Line 20: "the kids mean that their sale price does not compensate the costs generated" -compensate is the wrong word to use here.
Authors: OK. Please see the manuscript.
Line 25: It is poor English to state things to be higher as higher is referring to a level, instead recommend using other words to explain increase i.e. greater.
Authors: OK. Please see the manuscript.
Line 25: When discussing breeds it is better to say purebred rather than just pure.
Authors: OK. Please see the manuscript.
Line 25: doesn’t this equate to 27%
Authors: OK. Please see the manuscript.
Line 26: remove a after comma ", similar ADG"
Authors: OK. Please see the manuscript.
Line 26: space after 6 g
Authors: OK. Please see the manuscript.
Line 27: the ADG was not reduced as the lactation progressed, however the ADG was less and less as the lactation progressed.
Authors: OK. Please see the manuscript.
Line 28: "and milk energy value" – what is milk energy value?
Authors: OK. Please see the manuscript. A new sentence was written in "Materials and Methods" to explain how energy milk content was calculated.
Line 29: "that compensates said reduction" suggest changing to say "that will compensate for this reduced energy content".
Authors: OK. Please see the manuscript.
Line 30: "results show that 29 crossed kids " be more specific about what you are referring to ... "results show that MGxBoer crossbred kids" and "purebred MG kids".
Authors: OK. Please see the manuscript.
Line 30: "In conclusion, the results show that crossed kids reached the minimum slaughter weight a week earlier than pure kids, highlighting the need to provide a concentrated feed and the increased income from higher milk sales"…. The fact that the crossbred animals reached the slaughter weight earlier does not highlight the need to provide a concentrated feed. These are 2 factors, first, the crossbred kids reached the weight fastest, and another point you identified in the study is that the milk energy content is less. Based on these observations, in order to achieve the desired growth these goats should be provided with a concentrated feed. I would suggest you slightly change the
language from "a week earlier than pure kids, highlighting the need" to "a week earlier than purebred kids, and highlighted the need". Also, I don’t see how this highlights the increased income from higher milk sales?
Authors: OK. Please see the manuscript. A new sentence has been incorporated.
Line 32: key words "breeding system" – only that is not included in the title.
Authors: OK. Please see the manuscript.
Line 38-39: "which contemplates suckling and milking period postpartum and usually once-a-day milking, are two regular systems for the production of goat milk in Spain [2,3]"… this second system is difficult to understand... Also I don’t think contemplates is the correct word for this.. suggest writing "which includes a suckling period prior to the milking period, which is typically once-a-day milking.."
Authors: Please see the manuscript. The idea of a mixed rearing system consists of simultaneous suckling plus milking before weaning followed by the exclusive milking period.
Line 39: Change "are two regular" to "are the two main systems"
Authors: OK. Please see the manuscript.
Line 43: On the other hand,
Authors: OK. Please see the manuscript.
Line 44: "On the other, ARS presents difficulties due to investment in facilities and machinery [3], which leads to an increase in costs [6], and so maintains the interest of the MRS" this sentence doesn’t make sense… please re-write
Authors: OK. Please see the manuscript. The sentence has been changed.
Line 46: "kids is very low (120-168 g/day)". Compared to what?
Authors: OK. Please see the manuscript. A new sentence has been incorporated.
Line 47-48: " to mate females not used to obtain replacement with males of a meat breed" ... reword "to mate the females that are not used to produce the next generation of replacements, with males of a meat breed."
Authors: OK. Please see the manuscript.
Line 50: "average daily gain" …. Be consistent with abbreviations, so change to ADG.. and change "kids and that this will reduce" to "kids which will"
Authors: OK. Please see the manuscript.
Line 60: "MG breed goats (44 ± 2 kg)" … MG breed goats weighing on average 44 ± 2 kg…
Authors: OK. Please see the manuscript.
Line 66: raised one kid each?.. two kids each?
Authors: OK. Please see the manuscript.
Line 74: are the Ca and P per kg of DM too? as with the concentrate?
Authors: OK. Please see the manuscript.
Line 85-86: and adjusted, when necessary, to 35 day weights… why? And how? This has not been explained previously, why 35day?
Authors: OK. Please see the manuscript. The growth adjustment is necessary, for example, to determine the exact day on which the minimum slaughter weight is reached (discussion).
Line 88: what is actual milk yield? Was this daily milk yield? Weekly total milk yield? Total milk yield?
Authors: OK. Please see the manuscript. "Actual milk" is a term commonly used in this field, it refers to milked milk.
Line 94: change to "milk samples (50 ml) were collected..
Authors: OK. Please see the manuscript.
Line 101: "weighing method (WSW) in every week of the experiment, on Thursday" change to "weighing method (WSW) weekly."
Authors: OK. Please see the manuscript.
Line 110: evaluation?.. also "milk consumed by the kids"
Authors: OK. Please see the manuscript.
Line 111: milk characteristics? Which were?. Average daily gain of the kids or dams? also be consistent with abbreviations "ADG".
Authors: OK. Please see the manuscript.
Line 112: "with a repeated measures model. This mixed model included the fixed effects of.."
Authors: OK. Please see the manuscript.
Line 112: "genetic type" please specify.
Authors: OK. Please see the manuscript.
Line 113: "week of lactation and their interaction"? What is the week of lactation? Do you mean the week they kidded? Or the number of weeks they have been lactating for? What is the interaction with?
Authors: OK. Please see the manuscript.
Line 116: was analysed by a mixed model, but what software? Suggest saying "was analysed using the MIXED model procedure in SAS (reff)".
Authors: OK. Please see the manuscript.
Line 117: global average daily gain? Was this a global study?
Authors: OK. Please see the manuscript.
Line 126-127: change "experiment, so that the BW" to "experiment. The BW..", also, change "were significant on mortality" to "had a significant effect on kid mortality"
Authors: OK. Please see the manuscript.
Lines 128-129: not clear what the percentages are...
Authors: OK. Please see the manuscript.
Lines 130-131: "as the birthweight rose" - the birthweight doesn’t rise, it is one weight for each animal. However, some kids have greater/heavier BWs
Authors: OK. Please see the manuscript.
Line 132: "Thus, 100% of the kids with BW under "
Authors: OK. Please see the manuscript.
Line 139: crossbred kids
Authors: OK. Please see the manuscript.
Line 140: versus change to vs
Authors: OK. Please see the manuscript.
Line 141: Shouldn’t start sentence with an abbreviation.
Authors: OK. Please see the manuscript.
Line 142: was this ADG value the average ADG of the 100 kids?
Authors: OK. Please see the manuscript.
Lines 141-144: big sentence, suggest splitting.
Authors: OK. Please see the manuscript.
Lines 146: I wouldn’t say the evolution of ADG was negative, as they still grew during this time, the ADG trend however was decreasing as the individuals got older.
Authors: OK. Please see the manuscript.
Line 146-147: "the difference in ADG between… was greater…"
Authors: OK. Please see the manuscript.
Line 148: "period than at the end"
Authors: OK. Please see the manuscript.
Lines 148-149: please re-write as this doesn’t make sense.
The sentence is not relevant to the development of the manuscript and has been removed.
Line 164: "and sex factor". Maybe change wording.
Authors: OK. Please see the manuscript.
Table 3 – Name does not make sense. Also, it is difficult to understand the table. Are the values under 1 and 2 number of animals or milk yields? Also what does ESM stand for?
Authors: OK. Please see the manuscript.
Lines 170-171: "On the other hand, Table 4 shows that the average value of milk consumed by the kids evolved negatively as their age increased, a trend that is valid for the number of kids reared and for the genetic type"…. Doesn’t make sense, please reword... "The average consumption of milk decreased as the age of the kids increased (Table 4). This trend in milk consumed was consistent irrespective of the number of reared kids and the genetic type of the kids."
Authors: OK. Please see the manuscript.
Table 4- same comments as Table 3
Authors: OK. Please see the manuscript.
Line 180: the composition doesn’t depend on the stage of lactation, but it does change significantly based on the stage of lactation.
Authors: OK. Please see the manuscript.
Line 181-182: "in the later case" , do you mean the later stages of lactation?
Authors: OK. Please see the manuscript.
Line 183: what is milk energy? Never mentioned this before, how have you calculated this?
Authors: OK. Please see the manuscript. The description of the milk energy calculation has been incorporated into materials and methods
Lines 183-184: "Table 5 shows how the average values of milk energy tended to decrease from the second week to the sixth week" – if I’m reading the table correctly, the "milk energy" increases up to week 2 then decreased then increased then decreased then increased. …
Authors: OK. Please see the manuscript. A new sentence was written.
Table 5- similar comments as Table 3
Authors: OK. Please see the manuscript.
Line 191: "The level" the proportion? Mortality rate?
Authors: OK. Please see the manuscript.
Line 191: you compared the mortality to another paper, why could these mortality rates be different? Where was this other study conducted? Same breeds?
Authors: OK. Please see the manuscript. A new sentence was written.
Line 192: "the breeding process"? Please explain
Authors: OK. Please see the manuscript.
Line 194: abbreviation consistency - birth weight = BW
Authors: OK. Please see the manuscript.
Line 195: What do you mean by habitual mortality?
Authors: OK. Please see the manuscript. New sentence.
Line 199: "However, the BW of the MG kids is lower than the one obtained, in the same breed" the BW of the current study MG kids? Or what MG kids?
Authors: OK. Please see the manuscript.
Line 201: It is better to write crossbred rather than crossed. Also, these values are averages so you should say " the average BW of crossbred kids" etc.
Authors: OK. Please see the manuscript.
Line 201: "the BW of crossed kids is lower than the one reviewed by".. "the BW of crossbred kids are lighter than those reported by Goonerwardene et al. [17], who also evaluated the BW of crossbred kids from Boer sires and dairy breed dams such as Alpine (3950 g) and Saanen (4090 g). Although, .."
Authors: OK. Please see the manuscript.
Line 203: "all of them" – who is them?
Authors: OK. Please see the manuscript.
Line 204: "all these BW" – all of the BWs discussed…
Authors: OK. Please see the manuscript.
Lines 206-209: please re-word sentence, very hard to follow.
Authors: OK. Please see the manuscript. New sentence.
Lines 209-211: "Moreover, descending values of ADG with the progress of lactation observed in this work are contrary to what was reported by other authors." Suggest re-wording.. "Moreover, in this study the ADG reduced as the individuals got older, however this trend was not observed in other studies (reference the studies). For example, Vazquez..."
Authors: OK. Please see the manuscript.
Line 217: change raised to increased
Authors: OK. Please see the manuscript.
Line 219: "when raising the weight of their kids" – what do you mean by this?
Authors: OK. Please see the manuscript.
Lines 218-220: very confusing sentence, please re-word
Authors: OK. Please see the manuscript. New sentence.
Line 222: "these facts" – these observations
Authors: OK. Please see the manuscript. New sentence.
Line 223-224: for a hypothetical lower milk consumption" recommend saying lower before hypothetical.
Authors: OK. Please see the manuscript. New sentence
Lines 225: Refer to my comments about lines 183-184
Authors: OK. The sentence has been reworded.
Line 231: why especially the crossed animals?
Authors: OK. Please see the manuscript.
Line 232: with all this – new sentence so what are you referring to?
Authors: OK. The sentence has been reworded.
Line 232-233: "With all this, the final weight at 35 days of life of this work of pure MG kids (7293 g) was lower than that of Vazquez-Briz et al. [8] (8402 g), similar to that of Zurita-Herrera et al. [20] (7400 g) and higher than Pérez-Baena et al.
[7] (6560 g)." suggest re-wording "The average final weight of 35 day old purebred MG kids (7293 g) was similar to the 7400 g reported by Zurita-Herrera et al. [20], greater than Pérez-Baena et al. [7] (6560 g) but lower than that of Vazquez-Briz et al. [8] (8402 g)."
Authors: OK. Please see the manuscript.
Lines 234-236; also suggest re-wording this sentence to make sense e.g. "In Spain, purebred MG kids reach the minimum typical sales weight at 32 days, while the crossbred kids can achieve slaughter weight one week earlier, allowing an increase in the quantity of milk sold and improving the economic results of the farm."
Authors: The sentence has been reworded, although the sentence proposed by the referee has been modified to adapt it to the real situation.
Lines 238-241: Do you think these conditions may have had an effect on the results?
Authors: We think so, as the sanitary conditions of the udder negatively affect the composition of the milk and even the availability of the goats to be suckled.
Final comment, TSW was only briefly mentioned a few times despite this being one of the main objectives of the study. If you do not refer to this often then I would suggest not abbreviating.
Authors: OK. Please see the manuscript.
The authors wish to thank the reviewer for the proposals made

Reviewer 2 Report
This study provides some interesting results, but the manuscript is immature especially in Material and methods section. The following points should be substantially improved.
- The methodology did not provide information on how the mortality was analyzed - whether the mortality concerned only births or also falls during the experiment? There is also no information on the distribution of mortality from different pregnancies - single or twin.
- There is no information on the sex structure of kids in the analyzed breed groups (MG and MGxBoer), despite the fact that the study showed a significant influence of sex on the average daily gains (ADG).
- There is no information on how many MG and Boer rams were used in the experiment?
- The methodology describes - in detail - the analysis of the composition and quality of milk (e.g. the number of somatic cells), while the authors do not analyze these parameters in the text.
- References need to be supplemented.
Author Response
RESPONSE TO REVIEWER 2
To make the changes made in the manuscript more visible, these changes are written in red type.
Reviewer 2
Comments and Suggestions for Authors
This study provides some interesting results, but the manuscript is immature especially in Material and methods section. The following points should be substantially improved.
- The methodology did not provide information on how the mortality was analyzed - whether the mortality concerned only births or also falls during the experiment? There is also no information on the distribution of mortality from different pregnancies - single or twin.
Authors: OK. New information was introduced. Mortality rate did not take into account birth mortality. Please see the manuscript.
- There is no information on the sex structure of kids in the analyzed breed groups (MG and MGxBoer), despite the fact that the study showed a significant influence of sex on the average daily gains (ADG).
Authors: OK. New information was introduced. Please see the manuscript.
- There is no information on how many MG and Boer rams were used in the experiment?
Authors: OK. New information was introduced. Please see the manuscript.
- The methodology describes - in detail - the analysis of the composition and quality of milk (e.g. the number of somatic cells), while the authors do not analyze these parameters in the text
Authors: The composition of milk in fat, protein and lactose has been used to calculate the energy of milk and it helps to justify some of the results obtained from growing kids. The calculation of energy in milk has been made explicit in materials and methods. On the other hand, SCC serves to give an idea of the health status of the udder of the animals in the herd, which affects the composition of the milk and could even affect the behaviour of the mothers for suckling. The fact that the SCC values found in this experiment are lower than the mean values obtained for goats in a large study countrywide, seems to indicate that this experiment was not affected by these kinds of factors.
- References need to be supplemented.
Authors: OK. New references have been introduced.
Round 2
Reviewer 2 Report
My comments were taken into account sufficiently to allow the article to be accepted.